# Enhancing Interpretability in Medical Image Classification by Integrating Formal Concept Analysis with Convolutional Neural Networks

**DOI:** 10.3390/biomimetics9070421

**Published:** 2024-07-10

**Authors:** Minal Khatri, Yanbin Yin, Jitender Deogun

**Affiliations:** 1Department of Computer Science and Engineering, University of Nebraska-Lincoln, Lincoln, NE 68588, USA; mkhatri2@unl.edu; 2Department of Food Science and Technology, University of Nebraska-Lincoln, Lincoln, NE 68588, USA; yyin@unl.edu

**Keywords:** interpretability, FCA, CNN, histopathology, Warwick-QU, BreakHIS

## Abstract

In this study, we present a novel approach to enhancing the interpretability of medical image classification by integrating formal concept analysis (FCA) with convolutional neural networks (CNNs). While CNNs are increasingly applied in medical diagnoses, understanding their decision-making remains a challenge. Although visualization techniques like saliency maps offer insights into CNNs’ decision-making for individual images, they do not explicitly establish a relationship between the high-level features learned by CNNs and the class labels across entire dataset. To bridge this gap, we leverage the FCA framework as an image classification model, presenting a novel method for understanding the relationship between abstract features and class labels in medical imaging. Building on our previous work, which applied this method to the MNIST handwritten image dataset and demonstrated that the performance is comparable to CNNs, we extend our approach and evaluation to histopathological image datasets, including Warwick-QU and BreakHIS. Our results show that the FCA-based classifier offers comparable accuracy to deep neural classifiers while providing transparency into the classification process, an important factor in clinical decision-making.

## 1. Introduction

Cancer is a major health problem that affects millions of individuals worldwide. It is one of the leading causes of mortality in the world [1]. Advancements in digital technology have substantially contributed to the availability of histopathological image datasets. These images are high-resolution images of tissue samples obtained through the use of a microscope. In the field of cancer diagnosis, the examination of histopathological images can reveal whether a tumor is benign (non-cancerous) or malignant (cancerous). However, the traditional practice of relying solely on pathologists to examine these images is limited by their availability, experience, and risk of human error.

The application of machine learning techniques, including deep neural networks, for the classification of histopathological images is becoming increasingly common [2]. These computational techniques can automate the process of image classification, assisting pathologists in identifying cancerous tumors more accurately and potentially at an earlier stage.

In recent years, convolutional neural networks (CNNs) have emerged as a powerful tool for image classification tasks [3]. The most significant advantage of using a CNN is that it can automatically learn features from image data and handle non-linearity in the data. With the development of deep architectures for complex data, CNNs involve the computation and optimization of millions of parameters. Pretrained architectures such as VGG-16 [4], ResNet [5], and Inception [6] have made it feasible to work with smaller medical image datasets and have significantly improved the performance [7].

In the current literature, CNNs demonstrate promising results in classifying histopathological images as cancerous or benign. However, the reasoning behind the classification remains hidden, due to which, they are often treated as a black box model [8]. The lack of transparency of CNNs poses challenges in high-stake decision-making domains like healthcare, where interpretability is crucial for trust in the system.

Visualization tools like Saliency maps [9] can interpret the decision-making process of CNNs by highlighting the specific regions of an image that influence its classification. While such visualization helps in interpreting individual image classifications, it does not explicitly offer insights into relationships among multiple images based on shared high-level features, potentially deriving rules or patterns from these relationships. To bridge this gap, our research takes a two-fold approach: (1) extracting high-level, abstract features that have semantically rich representations of original images and (2) using the formal concept analysis (FCA) framework to identify relationships between these features and target classes. Our goal is to understand the relationships between multiple images based on these shared features, which could help us identify patterns or rules that make the decision process more transparent. While raw pixels provide granular information, high-level features extracted capture the semantics of images, and interpreting relationships at this level can be even more meaningful and insightful. This structured and relational interpretability is different from the pixel-wise importance interpretation provided by saliency maps for individual images. Using FCA on transformed feature space rather than raw pixel values can provide structured, rule-based insights, making it a valuable tool for enhancing the interpretability of classification.

The primary contribution of this paper is the development of an FCA-based classification model that enhances the interpretability of image classification for histopathological images. Our previous work [10] tested this approach on the MNIST handwritten digit dataset. The results were comparable to CNNs while providing interpretability. This study extends that work to histopathological image datasets, further validating the approach for complex applications.

## 2. Background

A recent survey [11] provides a comprehensive overview of machine learning (ML) and deep learning (DL) models used for the classification of colon cancer. The research suggests that DL model performs better than ML approaches when it comes to classifying image datasets of colon. Most of the studies utilize convolutional neural networks (CNNs) for the classification of histopathological images of the colon. Specifically, pretrained networks such as ResNet [12] and SqueezeNet [13] are used for classifying small datasets of colon cancer.

While CNNs have demonstrated promising results, it is important to consider the lack of transparency associated with this model. This becomes particularly relevant when applying CNNs for classifying histopathological images as either “cancerous” or “benign”. Previous studies have hypothesized the internal working of a CNN classifier in two steps: (1) feature extraction and (2) classification [5]. Now, we will discuss the techniques used for feature extraction and the models developed to add interpretability to classification.

### 2.1. Feature Extraction

Feature extraction is an important step in the analysis of image data, particularly for complex datasets like histopathological images. Histopathological images are highly detailed, containing a large number of features that capture the structural characteristics of the tissue. A large-dimensional dataset makes most of the classification methods infeasible due to high computational complexity. Therefore, feature extraction techniques are used to reduce the dimensionality of large datasets. Feature extraction transforms data from high-dimensional space to a relatively low-dimensional space by identifying important features and eliminating noise and redundancy. It allows the application of classification methods to the low-dimensional representation instead of the entire feature set. Overall, it helps to improve model performance and also speeds up the model training time.

The feature extraction methods vary from linear to non-linear methods.

Linear Methods: Linear transformation methods include Principal Component Analysis (PCA) [14] and Linear Discriminant Analysis (LDA) [15], which are widely used for dimensionality reduction. Principal Component Analysis (PCA) is an unsupervised technique which works by identifying the features (or axis) in the dataset that maximize the variance. The newly learned features are called “principal components” and are linearly independent.These components capture the major patterns and structures in the data in a hierarchical manner, from the most to the least important. In contrast, Linear Discriminant Analysis (LDA) is a supervised method which maximizes the separation between multiple classes. It does this by finding a linear combination of features that best separate data into different classes [14].Non-Linear Methods: To handle non-linear relationships, neural networks, particularly autoencoders, have gained attention. Autoencoders are a type of artificial neural network used for unsupervised learning. They can be used for various tasks such as dimensionality reduction, anomaly detection, and denoising. In addition, CNNs have potential in automatically learning features directly from raw image data. This has led to the development of advanced feature extraction architecture, including pretrained CNN models that can extract high-level features with accuracy. Medical image datasets are usually small because the procedure of collecting data is expensive. The limited availability of data is overcome by using pretrained CNN architectures like VGG16, ResNET, and Inception for feature extraction.

The selection of the feature extraction technique is based on type of data, available resources, and the problem we are trying to solve. In feature extraction, the extracted features are high-level representations derived from the original set, which can then be used for classification.

### 2.2. Interpretable Classification Models

In high-stake decision-making domains such as healthcare, the transparency of a model’s decision-making process is crucial. Clinicians and patients are more inclined to trust and rely on a model if they can gain insights into its reasoning mechanism. The consequences of errors in these domains can be misdiagnoses, which can potentially cause harm to patients in healthcare. As a result, there is an ongoing shift in research towards enhancing model interpretability along with the optimization of performance. This transition acknowledges that a model’s utility is not only determined by its accuracy but equally by its ability to provide clear insights into its decisions. As the complexities of models continue to grow, the challenge remains to strike a balance between performance and interpretability while ensuring that the models remain both powerful and transparent in their applications.

Machine learning models, such as decision trees [16], are employed in image classification due to their interpretable decision-making process that mirrors human reasoning. However, decision tree models have limitations. They are unstable to even small changes in input data, resulting in a completely different tree. They are also prone to overfitting, where the model can learn not only from the underlying patterns in the data but also from noise and outlier values within the features, which can degrade their performance on unseen data. To overcome the limitation of overfitting, an ensemble method like Random Forest [17] is used. While an individual decision tree is easy to interpret, the interpretation of results produced by Random Forest is not as straightforward.

As previously highlighted, convolutional neural networks (CNNs) have significantly advanced image classification tasks. However, by design, they are not interpretable. Several researchers have developed methods to add a level of transparency to the decision-making process. One study [18] combines the LIME (Local Interpretable Model-Agnostic Explanation) approach with a black box method to improve the understanding of the prediction of colon cancer. LIME is a technique that can explain individual predictions. In addition, techniques like saliency maps [9] and CAM (Class Activation Mapping) [19] are used to interpret the CNNs’ decision-making processes. While these enhance our understanding of the decision-making process, they provide local interpretations of the model and do not explicitly provide insight into the overall working of these models by establishing a relationship between the semantic or higher-level features that the model has recognized and the output labels. Therefore, in the next section, we will discuss the formal concept analysis framework, which shows potential in identifying these relationships.

### 2.3. Formal Concept Analysis (FCA)

Formal concept analysis (FCA) [20] is a mathematical framework for data analysis that focuses on identifying and visualizing relationships within a dataset. It is based on the notions of “concept” and “lattice” to form a structured representation of data. Concepts are defined as collections of objects that share common attributes, while a lattice is a visual representation of these concepts in a hierarchical manner to reveal the relationships, patterns, and outliers.

The input to formal concept analysis is a formal context, a triplet (G,M,I), where *G* is a set of objects, *M* is a set of attributes, and *I* is a binary relation between *G* and *M*, indicating which objects have which attributes. A formal context is represented as an incidence table, where rows represent a set of objects *G* and the columns represent a set of attributes *M*, with entries indicating the binary relationship. From this context, FCA generates a set of concepts. A formal concept is a pair (A,B) where the following definitions apply:*A* is a subset of objects or the extent of a concept.*B* is a subset of attributes or the intent of a concept.*A* is the set of all objects that share all the attributes in *B*, and *B* is the set of all attributes shared by all objects in *A*.

For example, a formal context of six images is represented in Table 1, where images are objects labeled as {1,2,3,4,5,6} and attributes are high-level features of images labeled as {a,b,c,d}. The “1” indicates the presence of a relationship, and “0” indicates the absence of a relationship. A concept with intent {a,b} has extent {2,3,4,5}.

Further, FCA generates a concept lattice which provides a visual representation of the hierarchy of all the concepts of a given context. It is represented by a line diagram called a “Hasse diagram”, as shown in Figure 1. The line diagram consists of nodes and all the objects and attributes are represented as labels. The nodes represent the concepts, and the intent and extent of each concept can be read from the line diagram by following the simple reading rule: The intent of a concept is the set of attributes on all the paths leading in the upward direction from the node and extent is the set of objects on all the paths leading in the downward direction from the node [20]. For example, in Figure 1, the node labeled “C2” is a concept with intent {a,b} and extent {2,3,4,5} and node labeled “C5” with intent {a,b,c} and extent {5}.

Concepts can have relationships with many other concepts; some concepts are more general than others. The relationship of generalization and specialization is modeled using the subconcept–superconcept relation. The extent of the subconcept is a subset of the extent of the superconcept. Likewise, the intent of the superconcept is a subset of the intent of the sub-concept [21]. In the concept lattice, when we move from top to bottom, we move from general to more specific concepts. In the Figure 1, the node labeled C5 is a subconcept of the node labeled C1, C2, and C3.

The advantages of using FCA as a classifier over a CNN are as follows:Abstract Feature Relationships: FCA creates clear mapping between images and their features, which is defined in the form of concepts. When applied to high-level features extracted, FCA can show relationships between images based on the shared abstract features they possess. FCA naturally leads to the derivation of if–then rules, which offer a human-readable explanation of why a particular decision was made.Hierarchical Understanding: The lattice structure organizes concepts hierarchically, which can provide insights into the structure of data and how different features relate to one another and share attributes among images.Incremental Model: FCA can handle changes in input data because it is an incremental model. The addition of new features or images can be incorporated by either adding a new concept or extending the existing concept without the need for retraining the model.Single Pass: FCA is efficient because it requires a single pass over the training dataset to construct the concept lattice.

In recent works, FCA is often used for clustering [22], association rule mining [23], and determining classification rules [24]. One survey [25] highlights the potential use of the formal concept analysis framework as a classifier in cancer diagnosis. In the next section, we discuss the details of our FCA-based classifier.

## 3. Methods

The proposed classifier consists of two main components, (1) feature extraction and (2) a formal concept analysis (FCA)-based classifier, as shown in Figure 2, which was used in our previous work [10] with a handwritten image dataset. In this section, we describe our approach and the histopathological image datasets used to evaluate the method.

In our proposed method, we use a feature extraction framework to identify the important features in the images and then use the extracted features to train our FCA model. Given that our FCA classifier could generate $2^{n}$ concepts (where *n* is the number of features) and FCA does not have the inherent ability to perform feature extraction, a pre-processing step is needed. The choice of feature extraction technique varies depending on the dataset used. Various feature extraction techniques like PCA, LDA, CNNs, and pretrained CNNs can be used. This step results in a set of high-level features learned from the original images.

The input to our FCA-based classifier is a formal context: an incidence table where rows represent the set of images from our dataset and the columns represent the binary features. To obtain the formal context, we first need to convert the multi-valued, high-level features obtained from the feature extraction step into a binary feature. This conversion involves two steps. First, we normalize these features to ensure uniformity across the dataset. Second, we set a threshold where every feature value above the threshold is assigned as “1”, indicating the presence of the feature, and “0” otherwise, indicating its absence. Same threshold is applied to all features across the dataset.

The classification approach based on formal concept analysis allows us to extract the relationship between high-level input features and the different output class labels based on the concepts discovered from data. Each concept represents groupings of images based on shared features. Conceptually, our classification method consists of a learning step and a classification step. In the learning step, we extract the relationships from a training set in the form of a concept lattice where we define a class label for each concept. In the classification step, we predict the class of instances in the testing set based on concepts in the training lattice.

We implement the FCA classifier algorithm as represnted in Algorithm 1 in Python using the concepts package to construct the concept lattice.
**Algorithm 1** FCA Classifier Algorithm1:**Input:** Training set $\left(X_{train},y_{train}\right)$, Testing set $\left(X_{test}\right)$2:**Output:** Predicted class labels for the testing set. 3:**Step 1: Constructing Training Lattice**4:Initialize an empty set $X_{t}$ and an empty list $y_{t}$5:**for** each feature vector $x_{i}$ in $X_{train}$ **do**6:   **if** $x_{i}\notin X_{t}$ **then**7:     Add $x_{i}$ to $X_{t}$8:     Add the corresponding class label $y_{i}$ to $y_{t}$ as a new list9:   **else**10:     Find the index of $x_{i}$ in $X_{t}$11:     Append the class label $y_{i}$ to the corresponding list in $y_{t}$12:   **end if**13:**end for**14:Construct the training lattice from the unique training feature set $X_{t}$ 15:**Step 2: Defining Class of a Concept in Training Lattice**16:**for** each concept in the training lattice **do**17:   Get the extent (set of images) of the concept18:   Count the number of images belonging to each class in the extent19:   Assign the class label with the maximum count as the class of the concept20:   In case of a tie, break the tie arbitrarily21:**end for** 22:**Step 3: Classify the Testing Set**23:Initialize an empty list $y_{pred}$ for storing predicted class labels24:**for** each feature vector $x_{j}$ in $X_{test}$ **do**25:   Calculate the Hamming distance between $x_{j}$ and the intent of each concept in the training lattice26:   Identify the concept with the smallest Hamming distance27:   Assign the class of the identified concept as the predicted class label for $x_{j}$28:   Append the predicted class label to $y_{pred}$29:   In case of a tie in the Hamming distance, break the tie arbitrarily30:**end for** 31:**return** $y_{pred}$

We use “accuracy” to evaluate the performance of our proposed classifier. Moreover, the lattice diagram helps in understanding the hierarchical learning and relationship between features and class labels.

Our proposed image classifier is a generic framework which can be applied to all image datasets. However, for different datasets, we can modify the feature extraction framework by incorporating new techniques that improve the classification result. In this study, we applied the proposed classifier on two well-known histopathological image datasets, Warwick-QU and BreakHIS dataset.

## 4. Results and Discussion

The Warwick-QU dataset [26] (Figure 3b) comprises 165 images with 74 benign and 91 malignant colon tissue images. Each image is 522×775 pixels in size. We used an approximate 75–25% train–test split with equal numbers of benign and malignant images in the test set. The BreakHis Dataset [27] (Figure 3a) comprises 7909 histopathological images with different magnifications, including 40x, 100x, 200x, and 400x, collected from 82 patients. There are 2480 benign tumor images and 5429 malignant tumor images.

For the Warwick dataset, we explored different convolutional neural networks (CNNs) for feature extraction: (1) ResNet50 [28]: ResNet50 is a deep convolutional network consisting of 50 layers, known for its “residual blocks”, which help in training deeper networks. (3) VGG16 [29]: VGG16 has 16 layers, while VGG19 has 19 layers (16 convolutional and 3 fully connected layers). (4) Inception v3 network [30]: It is a convolutional neural network with several layers of convolution, pooling, and fully connected layers.

Although these CNNs were pretrained on the ImageNet dataset, they were not specifically trained on histopathological images. Given the small size of the Warwick dataset, we leveraged the transfer learning approach to enable the models to learn relevant features specific to histopathology. The idea behind transfer learning is to leverage knowledge gained while solving one problem and apply it to a different but related problem. The BreakHis dataset, with its breast cancer histopathological images, shares similarities with the Warwick-QU dataset (Figure 3). Additionally, BreakHis is much larger than the Warwick dataset. We used global normalization to normalize both datasets to ensure consistency.

First, we pretrained the CNNs on the BreakHis dataset; then, we fine-tuned the models using the Warwick training set. Finally, we tested the classifier on the Warwick test set to evaluate performance.

For each of these CNNs, we added a fully connected dense layer on top of the pretrained model. We experimented with different numbers of neurons in the fully connected layer and the number of layers, aiming to keep the number of features low as the training time of our FCA model depended on the number of features. The final architecture of the dense layer included three layers with 128, 32, and 16 neurons, respectively. The output from the last dense layer, consisting of 16 neurons, was used as the feature vector, which was then passed to the FCA classifier.

We compared the classification performance of the FCA classifier for each feature extraction technique pretrained on the BreakHis dataset and fine-tuned on the Warwick training dataset. The performance metrics for the FCA classifier are reported in Table 2. For each model, the number of features extracted from the CNN model was 16. By pretraining on BreakHis, our model improved at recognizing cancer-related features, which were then fine-tuned on the Warwick dataset. By leveraging these feature extraction techniques, our aim was to optimize the FCA classifier’s accuracy.

Additionally, we evaluated the performance of the CNN models when used directly as classifiers, where the fully connected dense layer is followed by a classification layer. These results are presented in Table 3.

The results in Table 2 show that the Inception v3 model, when used for feature extraction, and FCA as a classifier achieved the highest accuracy of 95.23%, with sensitivity of 100% and specificity of 81%. Table 3 shows the performance of the CNN models when used as classifiers themselves. Inception v3 showed the best performance with accuracy of 80.95%, followed by VGG16 with accuracy of 78.57% and ResNet 50 with accuracy of 61.90%. These results indicate that while all models benefited from the transfer learning approach, the FCA classifier combined with Inception v3 features significantly outperformed the models used directly as classifiers.

To show the importance of concept lattices in visualizing the relationship between high-level features extracted from CNNs and the output class labels, we present the concept lattice for the Inception v3 model used for feature extraction, which achieved FCA classification accuracy of 90.4%. This is represented in Figure 4 for simplicity. Each node in the lattice represents a concept with a specific set of features (intent), set of images (extent), and a class label (0 or 1, where 0 means benign and 1 means malignant). The set of abstract features is {a,b,c,d,e,f,g,h,i,j,k,l,m,n,o,p}, and concepts are labeled either 0 or 1, which represent class labels for a binary classification problem. For simplicity, we did not represent the extent for the concepts in the lattice. For instance, the concept c6 is associated with the feature {a,f,m} and the class label 1. This lattice helps in identifying the important features or combinations of features that are strong indicators of a class and identifying patterns or rules that can be directly applied in decision-making processes.

Predominant Features for a Class:For Class 1: Features on the right side of the lattice, {a,k,l,e,g,c}, are predominantly associated with class 1. This suggests that the presence of these features in an image makes it more likely to be classified as class 1.For Class 0: The left side with features, {j,o,h,p}, has nodes predominantly labeled 0, suggesting images with these features are more likely to be classified as class 0.Some nodes have multiple features associated with them, indicating those features frequently occur together in the dataset. For example, concept c2 is associated with features *j* and *o*, suggesting that these two features tend to co-occur. The combination of *j* and *o* (concept c2) is exclusively associated with class 0. This can be a strong rule in the classifier.If two features never appear together in a along a path from subconcept to superconcept, they might be mutually exclusive or at least rarely co-occur in a dataset. For instance, there is no direct path connecting the left features, {h,p,j,o}, and the right features, {a,k,l,e,g,c}, suggesting they do not often appear together in the same instances.Hierarchy of Concepts:The lattice shows a clear hierarchical relationship between concepts, with more general concepts at the top. As we move down the lattice, the concepts represent a more specific combination of features. For instance, c1 and c3 are more general concepts at the top, while c10, c14, and c15 are among the most specific concepts at the bottom. Concept c15 does not have a class label defined because its extent has no images. The hierarchy can reveal which features often occur together. For example, the path from c6 (with feature *a*) to c3 (with feature *f*) suggests that in the dataset, when *a* is present, *f* is also likely present.Class Inheritance:If a superconcept has a class label, its subconcepts will generally have the same label unless specified otherwise. For example, if a superconcept is labeled 1, and there is no contradicting information in the subconcept, we can infer the subconcept is also class 1. In the lattice, c6 with feature *a* inherits the class label 1 from c3 with feature *f*, as well as concepts c9, c11, c12, c13, and c14.Some features might appear in both class 0 and class 1 concepts, making them ambiguous in terms of classification. By looking at the hierarchy, we can determine if a feature’s classification power changes depending on its combinations with other features. For example, concept c2 has class label 0, and c3 has class label 1, while for concept c5, sharing the intent from both c2 and c3 makes it stronger towards class 0.

We compared the performance of our proposed FCA classifier with the established benchmark method on the Warwick dataset from a study using the ResNet50 model [12] and the Inception v3 classifier, which was used in the feature extraction step. The comparison results are shown in Table 4. In the existing literature, the highest accuracy on the Warwick dataset is 88% using the ResNet50 network [12] . Our model achieved accuracy of 95.23 % with the same train–test split of 75–25% as reported in the study. In addition, we compared the performance of the FCA classifier with the Inception v3 classifier, where it achieved significantly lower accuracy of 80.95 % Therefore, in the Warwick dataset, the FCA-based classifier shows promising results.

To further validate and test the applicability of the FCA model on other histopathological images, we tested it on the BreakHIS dataset. The accuracy is reported for each magnification on the FCA classifier as well as the Inceptionv3 classifier, which is used for feature extraction for the FCA classifier and has the highest accuracy reported from an SVM classifier using DenseNet201 (CNN) for extracting features in the current literature [31], as shown in Table 5.

Further, to validate the performance, we used cross-dataset performance. We first used the BreakHIS dataset to train the Inception v3 model and then used the trained model to extract features for the entire Warwick dataset and used these features as input to the FCA classifier for classification on the Warwick dataset. The results are reported in Table 6. Here, the model did not use the Warwick dataset in the feature extraction step to learn the features.

Overall, the results show that the FCA-based classifier significantly improves both interpretability and accuracy in medical image classification. For smaller datasets like Warwick-QU, transfer learning using pretrained architectures such as Inception v3 helps to improve classification accuracy. With a classification accuracy of 95.23% on the Warwick-QU dataset, the FCA-based model outperformed the existing benchmark. Furthermore, while CNNs have slightly better accuracy on the BreakHIS dataset, our FCA model provides enhanced interpretability, highlighting the generalizability and robustness of our approach across various histopathological image datasets of different sizes. Future research should focus on expanding the evaluation to more diverse datasets, improving scalability and exploring its utility in real-time medical diagnostic applications.

## 5. Conclusions

This study presents a novel image classification approach that integrates formal concept analysis (FCA) with convolutional neural networks (CNNs) to enhance the interpretability of image classification processes in medical imaging. The developed methodology utilizes the strengths of FCA to add a layer of transparency to the classification process by explicitly identifying relationships between high-level (abstract) features and class labels; the FCA-based classifier thus enhances both the accuracy and interpretability of model predictions. The application of FCA allows for the visualization of how features correlate with classification outcomes, addressing a critical gap in CNN-based classifiers that often operate as “black boxes”. The ability of FCA to systematically organize and interpret complex feature interactions enhances trust and reliability in automated medical diagnostic systems. While our findings are promising, achieving the optimal balance between accuracy and interpretability remains a challenge. The advancements made in this study can direct the future research in the following directions: applying FCA classifiers to diverse image datasets and testing their efficacy across various domains. Another key area will be to investigate the application of FCA in environments where data are continuously updated. Such application is crucial to determine the model’s robustness and adaptability in real-world situations. Additionally, future research should focus on integrating visualization methods such as Grad-CAM or Guided Grad-CAM with the FCA framework. These methods can help visualize the regions where the CNN pays attention during classification, enhancing the interpretability of the FCA classifier at the input level.

## Figures and Tables

**Figure 1 biomimetics-09-00421-f001:**
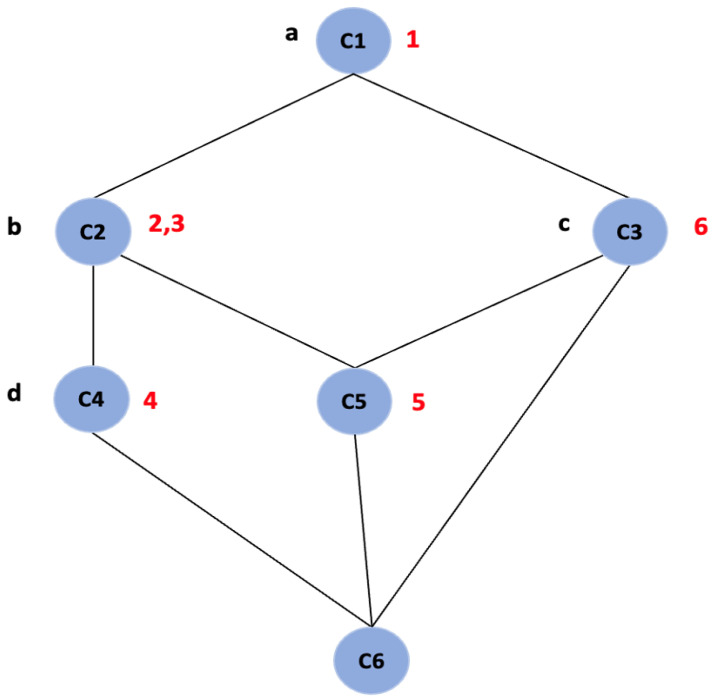
Concept Lattice of formal context of images.

**Figure 2 biomimetics-09-00421-f002:**
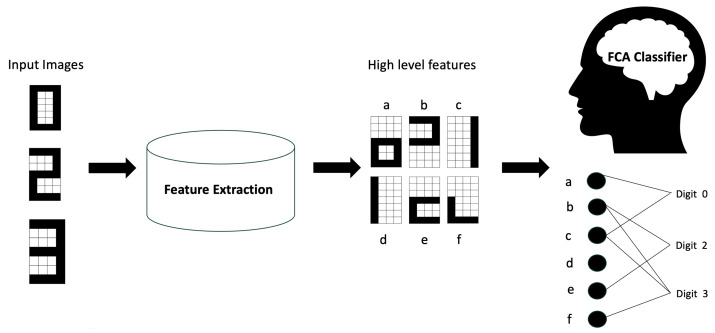
Workflow of the proposed FCA-based classification approach: In the initial step, input images (e.g., digit images) are processed using feature extraction methods to obtain high-level features, represented here as a,b,c,d, e, and f. These features are input to the FCA classifier which identify the relationship between the high-level features and the class labels, represented here as Digit 0, Digit 2, and Digit 3.

**Figure 3 biomimetics-09-00421-f003:**
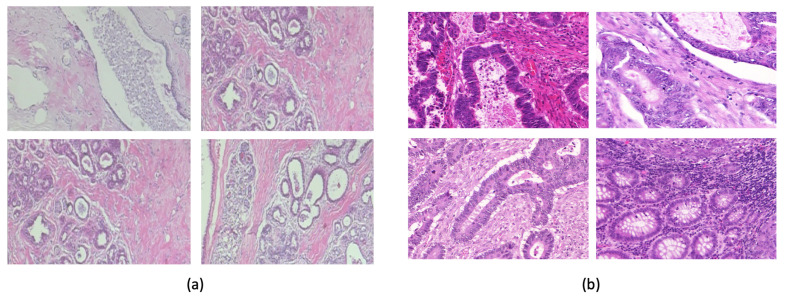
(**a**) BreakHis: Breast cancer histopathological images. (**b**) Warwick QU dataset: Colon cancer histopathological images. The scale bar represents X units.

**Figure 4 biomimetics-09-00421-f004:**
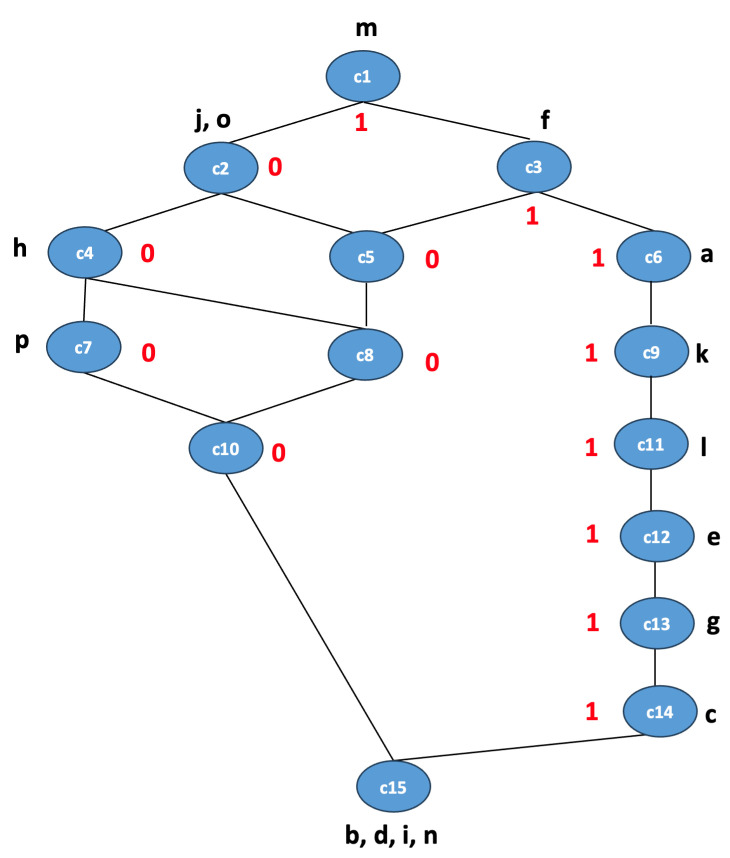
Concept lattice for the Warwick dataset. {c1,c2,…,c15} represents concepts, {a,b,…,p} represents features, and {0,1} represents class labels.

**Table 1 biomimetics-09-00421-t001:** Formal context of images.

Objects	a	b	c	d
1	1	0	0	0
2	1	1	0	0
3	1	1	0	0
4	1	1	0	1
5	1	1	1	0
6	1	0	1	0

**Table 2 biomimetics-09-00421-t002:** Comparison of feature extraction models pretrained on BreakHIS dataset and FCA-based classifier performance on Warwick dataset.

Feature Extraction	Accuracy %	Sensitivity %	Specificity %	F1 Score %
ResNet 50	69.04	90	43	73
VGG 16	80.95	90	71	83
Inception v3	95.23	100	81	91

**Table 3 biomimetics-09-00421-t003:** Comparison of CNN models pretrained on BreakHIS dataset used as classifiers on Warwick dataset.

Model	Accuracy %	Sensitivity %	Specificity %	F1 Score %
ResNet 50	61.90	71	52	65
VGG 16	78.57	90	67	81
Inception v3	80.95	95	67	83

**Table 4 biomimetics-09-00421-t004:** Comparison of performance on Warwick-QU dataset.

Classifier	Accuracy %	Sensitivity %	Specificity %
FCA	95.23	100	81
Sarwinda et al. [12]	88	89	87
Inception v3	80.95	95	67

**Table 5 biomimetics-09-00421-t005:** Comparison of accuracy on BreakHis dataset.

Magnification	FCA	Inception v3	Hao et al. [31]
40x	95.98%	94.49%	96.75%
100x	94.96%	94.72%	95.21%
200x	96.27%	95.04%	96.57%
400x	97.76%	88.19%	93.15%

**Table 6 biomimetics-09-00421-t006:** Results for cross-dataset evaluation. Models were trained on BreakHIS dataset and tested on Warwick dataset.

Classifier	Accuracy %	Sensitivity %	Specificity %	F1 Score %
FCA	81.21	92	66	84
Inception v3	69.09	49	93	64

## Data Availability

Data supporting the reported results were obtained from publicly available datasets, BreakHIS dataset (https://www.kaggle.com/datasets/ambarish/breakhis, accessed on 19 May 2024) and Warwick dataset (https://warwick.ac.uk/fac/cross_fac/tia/data/glascontest/download, accessed on 19 May 2024).

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
