# Peer review of "Enhancing Interpretability in Medical Image Classification by Integrating Formal Concept Analysis with Convolutional Neural Networks"

_biomimetics, 2024, doi:10.3390/biomimetics9070421_

Round 1
Reviewer 1 Report
Comments and Suggestions for Authors
In this paper, authors proposed binary classification method for histopathology images using formal concept analysis. The idea can be novel; however, authors must clarify several issues to help readers understand their proposal.
1. Authors use several CNN, such as ResNet50, VGG16 and Inception v3 to extract features from the histopathology images to use as attributes in Formal Concept Analysis. I understand that feature extracted from some CNN is feature map produced by almost last layer, such as fc6 or fc7. Which is the dimension of the feature vector?
2. Authors mentioned in line 283, “The concept lattice for the best model is represented in Figure4”. I understand that the best model is Pretrained Inception v3 on BreakHIS according to table 2. The concept lattice obtained from this best model applying Warwick Dataset is shown by Figure 4. From this figure, the number of extracted features is only 16.
(1) How to extract these 16 features from Inception v3?
(2) Does the number of features differ depending on the CNN model?
(3) Is it possible to visualize these features? For example, the most active neuron in the feature map.
3. In table 4, authors mentioned pre-trained CNN. There are several CNNs, please specify.
4. The proposed method is binary classifier: malignant or benign. This method can be applied for multiclass classification. If the response is yes, please explain.
5. Please comment on some limitations of the FCA-based classifier?
6. In the classification or diagnostic in medical images, besides accuracy, other metrics such as sensitivity, specificity and confusion matrix are important. Please add these metric values in the comparison table.
7. The visualization method, such as Grad-Cam or Guided Gram-Cam indicates regions where CNN put attention to make it decision of classification. Using the proposed FCA method this type of visualization can be possible?
8. I found some English errors. Please fix them.
Author Response
Thank you very much for taking the time to review this manuscript. Please see the point-by-point response attached.

Reviewer 2 Report
Comments and Suggestions for Authors
In this study, the authors proposed a novel deep learning-based framework for histopathological image classification. Specifically, the authors explored the integration of various feature extraction methods and formal concept analysis for cancer diagnosis, utilizing two publicly available datasets covering colon and breast cancer. Overall, the study is well-designed, and the methods are clearly explained. This study is also timely, given the significant increase in the volume of digital pathology data, while computational methods to effectively analyze these data are relatively lacking. However, I still have the following concerns that I wish the authors could address:
-
It is interesting to see that the transfer learning strategy could further improve prediction accuracy. However, I am wondering if the authors performed any color normalization in their pre-processing steps. Given that the two datasets are vastly different in color space, as shown in Figure 3, I am surprised to see this improvement. If not, I would encourage the authors to perform additional color normalization to see if this could further refine the performance.
-
I understand that the primary goal of this project is to justify the usage of Formal Concept Analysis for digital pathology tasks such as image classification. However, it is still desirable to see how much the model can improve compared to the baseline. The authors should at least use canonical methods such as directly using ResNet50 for classification and then compare with the proposed method or include state-of-the-art performance using the same datasets.
- I would expect the authors to include an independent dataset for validation.
Author Response

(The authors gave the same response as above.)

Round 2
Reviewer 1 Report
Comments and Suggestions for Authors
The authors attended correctly to my observations, and the revised version of the paper is very clear about the principal idea and its contribution. So, I am satisfied with this revised version. I recommend the publication of this paper.
Reviewer 2 Report
Comments and Suggestions for Authors
The authors have fully addressed my concerns. I do not have further comments.